# Rno-miR-130b Attenuates Lipid Accumulation Through Promoting Apoptosis and Inhibiting Differentiation in Rat Intramuscular Adipocytes

**DOI:** 10.3390/ijms26041399

**Published:** 2025-02-07

**Authors:** Yichen Yu, Yongfang Chen, Lijun Wang, Ji Cheng, Min Du, Shifeng Pan

**Affiliations:** 1Guangling College, Yangzhou University, Yangzhou 225009, China; yuyichen1608@163.com; 2College of Veterinary Medicine, Yangzhou University, Yangzhou 225009, China; chenyongfang77@163.com (Y.C.); wlj6379@163.com (L.W.); 008743@yzu.edu.cn (J.C.); 3Department of Animal Sciences, Washington State University, Pullman, WA 99163, USA; min.du@wsu.edu; 4Jiangsu Co-Innovation Center for Prevention and Control of Important Animal Infectious Diseases and Zoonoses, Yangzhou University, Yangzhou 225009, China

**Keywords:** rno-miR-130b, cell proliferation, apoptosis, adipogenic differentiation, rat intramuscular preadipocytes

## Abstract

Our previous studies have shown that miR-130b can significantly inhibit subcutaneous fat deposition in pigs. This study aims to further investigate its effect on lipid accumulation at early-stage (24 and 48 h) and late-stage (7 d) adipogenic differentiation and to clarify potential mechanisms using primary rat intramuscular preadipocytes (IMAs). Results showed that at 24 h and 48 h, miR-130b overexpression significantly reduced lipid deposition by inhibiting proliferation and inducing apoptosis. Furthermore, miR-130b overexpression significantly inhibited the expression of cell cycle and apoptosis marker genes. Specifically, the mRNA expression of *Ccnd1* tended to decrease, while the BCL2 protein level was significantly decreased at 48 h. In contrast, miR-130b inhibition significantly increased the BCL2 protein level. At 7 d, the miR-130b mimic significantly decreased intracellular TG content and tended to decrease *Hsd11b1* mRNA expression while significantly promoting *Lpl* mRNA expression. Additionally, the miR-130b mimic significantly increased the CASP3 protein level and tended to decrease the BCL2 protein level. In conclusion, our data indicated for the first time that miR-130b could reduce lipid deposition in rat IMAs through different mechanisms: at the early stage of differentiation by inhibiting proliferation and promoting apoptosis and at the late stage by inhibiting adipogenic differentiation, promoting lipid hydrolysis, and promoting apoptosis.

## 1. Introduction

Intramuscular fat (IMF) and subcutaneous fat (SCF) contents are crucial factors in determining the economics of meat quality in livestock and poultry. Therefore, inducing IMF and reducing SCF content are reasonable and effective methods to meet the growing demand for high-quality meat products. IMF refers to the lipid stored in skeletal muscle tissue, which is considered to be one of the most important factors determining meat quality [1,2]. The deposition of IMF is an extremely complex physiological and biochemical process, including the proliferation and differentiation of preadipocytes, lipid synthesis and decomposition, and fatty acid metabolism and transport [3]. Among these, the number and size of intramuscular adipocytes are the primary factors that regulate IMF content [2]. At the cellular level, lipid accumulation is mainly manifested by an increase in the preadipocyte and adipocyte number (hyperplasia) and adipocyte size (hypertrophy). At the molecular level, it mainly depends on the complex and accurate regulation of adipogenesis-related genes, transcription factors, and epigenetic mechanisms. The adipogenic differentiation of intramuscular preadipocytes (IMAs) into mature adipocytes mainly involves peroxisome proliferator-activated receptors (*PPAR*s), the CCAAT enhancer binding proteins (*CEBPs*) family, and other key adipogenic genes [4]. Besides multiple key genes for adipogenesis, intramuscular lipid accumulation is also closely related to the regulation of several classical signaling pathways [5].

MicroRNAs (miRNAs) are defined as endogenous non-coding RNAs that are highly conserved in different species, playing a role in RNA silencing by repression or degradation of target genes at the post-transcriptional level [6]. As a crucial epigenetic regulator, miRNAs play significant roles in regulating lipid deposition. In previous studies, a large number of microRNAs have been identified, and their functions in the process of lipogenesis or lipid metabolism have been demonstrated [7]. A previous study found that the expression of miR-17-92 significantly increased during the adipogenic differentiation of 3T3-L1 cells, and its overexpression could significantly induce adipogenic differentiation and increase triglyceride (TG) accumulation [8]. Furthermore, mice deficient in miR-206 showed increased adipogenesis following muscle injury, while overexpression of miR-206 abrogated adipogenic differentiation of fibro/adipogenic progenitors (FAPs) by targeting the transcription factor RUNX1, which uncovered a molecular fate switch, involving miR-206 and RUNX1, that controls FAP differentiation to adipocytes [9]. In addition, miR-15a/b was able to promote adipogenesis in porcine preadipocyte by inhibiting FOXO1 expression [10]. Moreover, miRNA-23a has been considered as a new type of adipogenic depressor that will play an important role in regulating adipocyte differentiation, since overexpression of miRNA-23a significantly decreased lipid accumulation and TG content in 3T3-L1 adipocytes by decreasing mRNA levels of adipocyte-specific genes involved in lipogenic transcription, fatty acid synthesis, and fatty acid transport [11]. In addition, further studies have also shown that circulating miRNAs in animals are mainly derived from adipocytes [12]. Therefore, all of the above results have clearly demonstrated that there is an obvious programmed regulatory relationship between miRNA and adipocytes.

Mounting evidence shows that miR-130b is involved in the regulation of a variety of biological processes. Our previous study suggested that miR-130b can significantly inhibit lipid deposition in porcine preadipocytes derived from subcutaneous adipose tissue by repressing peroxisome proliferator-activated receptor gamma (PPARG) expression [13], and our in vivo study also confirmed the same effect on porcine fat deposition [14]. In addition, we further demonstrated that microvesicle-shuttled miR-130b reduced fat deposition in recipient primary cultured porcine adipocytes by inhibiting PPARG expression, and intravenous injection of microvesicle-delivery miR-130b alleviated high-fat diet (HFD)-induced obesity in C57BL/6 mice through translational repression of *Pparg* [15]. Meanwhile, a previous study identified the miR-130b/301b cluster as a new regulator that suppresses beige adipogenesis involving PPARCG1A and PRKAA2 signaling in inguinal white adipose tissue (WAT) and is therefore a potential therapeutic target against obesity and related metabolic disorders [16]. Many studies have also found that miR-130b is significantly upregulated in gastric cancer tissues, and miR-130b can significantly increase the activity of the cancer cells [17]. In addition, the expression of miR-130b in colorectal cancer (CRC) tissues is also significantly increased, but in vitro studies found that miR-130b overexpression significantly inhibited the proliferation of CRC cells and promoted apoptosis. miR-130b also proved to be able to inhibit the growth of CRC tumors in vivo [18], suggesting that miR-130b plays a vital role in regulating cell proliferation and apoptosis. However, whether miR-130b inhibits lipid deposition in rat IMAs through regulation of cell proliferation and apoptosis still needs to be further investigated.

In this study, using rat IMAs as a cell model, we further explored the regulatory effects of miR-130b on cell proliferation, cell cycle, apoptosis, and lipid deposition in rat IMAs to provide a theoretical basis for elucidation of the mechanism of miR-130b inhibiting lipid deposition by regulating IMA proliferation and apoptosis in rats.

## 2. Results

### 2.1. Overexpression of MiR-130b Significantly Inhibited Lipid Deposition in Rat IMAs

After miR-130b mimic transfection, adipogenic differentiation was induced by the “cocktail method” for 24 h and 48 h. Lipid droplet deposition was observed under a microscope, and both oil red O staining extraction and a TG quantitative detection kit were used to comprehensively investigate the effect of miR-130b overexpression on lipid deposition in IMAs. The results showed that overexpression of miR-130b significantly inhibited the number of oil red O-stained positive cells in rat IMAs and significantly reduced lipid deposition at both 24 h (*p* < 0.01) and 48 h (*p* < 0.01) (Figure 1A). In contrast, compared with the iNC control group, miR-130b inhibitor treatment significantly increased the number of positive cells and lipid content at 24 h after IMA adipogenic differentiation (*p* < 0.01). However, there was no significant difference in TG content between the miR-130b inhibitor group and iNC group at 48 h (Figure 1B).

### 2.2. Overexpression of MiR-130b Significantly Promoted the Apoptosis of Rat IMAs

Flow cytometry was used to evaluate the changes in rat IMA apoptosis levels after miR-130b overexpression and differentiation induction for 48 h. The results showed that compared with the mNC group, both the late apoptosis level (*p* = 0.091) and the total apoptosis level (*p* = 0.056) tended to increase in the miR-130b overexpression group (Figure 2A), suggesting that the miR-130b mimic was involved in the early stage of adipogenic differentiation in rat IMAs and finally inhibited lipid deposition. However, miR-130b inhibitor treatment and differentiation induction for 48 h significantly reduced the levels of late apoptosis (*p* < 0.05) and total apoptosis (*p* < 0.01) compared with the iNC group (Figure 2B), suggesting that miR-130b inhibition could reduce the apoptosis level of rat IMAs and significantly increase the lipid deposition in the early stage of adipogenic differentiation of rat IMAs.

### 2.3. Overexpression of MiR-130b Significantly Inhibited the Proliferation of Rat IMAs

The effect of miR-130b overexpression on the proliferation of rat IMAs after adipogenic differentiation for 24 h and 48 h was detected by the EdU method. The results showed that compared with the mNC group, the miR-130b mimic group showed significantly reduced amounts of green fluorescence and fluorescence intensity after adipogenic differentiation for both 24 h (Figure 3A) and 48 h (Figure 3B). Further statistical analysis using Image J software showed that the number of EdU-stained positive cells was significantly reduced in the miR-130b mimic group compared with that of the mNC group (*p* < 0.01), indicating that the miR-130b mimic was able to significantly inhibit the proliferation of rat IMAs. In addition, compared with the iNC group, miR-130b inhibition significantly enhanced the intensity of intracellular green fluorescence after adipogenic differentiation for 24 h (*p* < 0.01) (Figure 3C), suggesting that the cell proliferation level was significantly increased. However, there was no significant difference between the miR-130b inhibitor group and iNC group after adipogenic differentiation for 48 h (Figure 3D). These above results suggest that miR-130b inhibition could significantly promote cell proliferation after adipogenic differentiation for 24 h.

### 2.4. Overexpression of MiR-130b Changed Cell Cycle Phases at Different Time Points in Rat IMAs

Flow cytometry was used to detect the effect of miR-130b overexpression on the cell cycle arrest of rat IMAs after adipogenic differentiation for 48 h. The results showed that there was no significant change in the G0/G1 phase between the miR-130b mimic group and mNC group, while the S+G2/M phases were significantly increased in the miR-130b mimic group (*p* < 0.05) (Figure 4A), suggesting that the cell cycle of rat IMAs was arrested in the S and G2/M stages by the miR-130b mimic. In addition, compared with the iNC group, miR-130b inhibition significantly reduced the G0/G1 phase of rat IMAs (*p* < 0.05) but had no significant effect on the S+G2/M phases (Figure 4B). These above results suggest that miR-130b overexpression could inhibit cell cycle progression of rat IMAs and thus inhibit cell proliferation.

### 2.5. Overexpression of MiR-130b Promoted Apoptosis by Inhibiting BCL2 Expression in Rat IMAs

The expression of factors related to adipogenic differentiation and apoptosis in rat IMAs after adipogenic differentiation for 48 h was determined by RT-qPCR and Western blot. Results showed that compared with the mNC group, the mRNA expression of B-cell lymphoma 2 (*Bcl2*) had no change in the miR-130b mimic group in rat IMAs after adipogenic differentiation for 48 h, while that of caspase 3 (*Casp3*) was significantly decreased (*p* < 0.05) (Figure 5A,B). In addition, the mRNA level of cyclin D1 (*Ccnd1*) tended to decrease in the miR-130b mimic group compared with the mNC group (*p* = 0.078) (Figure 5C), suggesting that miR-130b overexpression could induce rat IMA cycle arrest by inhibiting *Ccnd1* expression. Western blot results showed that the protein level of anti-apoptotic factor BCL2 was significantly decreased in the miR-130b mimic group (*p* < 0.001), while that of adipogenic markers PPARG and sterol regulatory element binding transcription factor 1 (SREBF1) were not significantly different (Figure 5D–H). These above results suggest that the miR-130b mimic could induce the apoptosis of rat IMAs by promoting *Casp3* expression and inhibit proliferation by repressing *Ccnd1* expression.

### 2.6. MiR-130b Inhibition Repressed Apoptosis by Increasing BCL2 Expression in Rat IMAs

In addition, we further determined the expression of factors related to proliferation, adipogenic differentiation, and apoptosis after miR-130b inhibition, and our results showed that compared with the iNC group, no significant difference was observed in the mRNA expression of *Bcl2*, *Ccnd1*, and *Casp3* in the miR-130b inhibitor group (Figure 6A–C). Western blot results showed that the BCL2 protein level was significantly increased in the miR-130b inhibitor group (*p* < 0.001), while the protein expression of CASP3 and PPARG showed no change. Interestingly, that of SREBF1 showed an upward trend (*p* = 0.086) (Figure 6D–H). These above results suggest that miR-130b inhibition could repress apoptosis by increasing BCL2 expression in rat IMAs.

### 2.7. Overexpression of MiR-130b Promoted Apoptosis by Inhibiting BCL2 and Increasing CASP3 Expression in Rat IMAs

In order to determine the effect of miR-130b on adipogenesis, lipid hydrolysis, and apoptosis in rat IMAs at the late stage of differentiation (7 d), we further detected the expression of genes related to the above processes. Results showed that compared with the mNC group, miR-130b mimic treatment significantly increased miR-130b expression (*p* < 0.01) and reduced both the number of oil red O-stained positive cells and the intracellular TG content (*p* < 0.05) in rat IMAs after adipogenic differentiation for 7 d. In addition, compared with the iNC group, miR-130b inhibitor treatment significantly decreased miR-130b expression (*p* < 0.01) and slightly increased TG content (Figure 7A–D), suggesting that miR-130b overexpression obviously inhibited the lipid accumulation of IMAs at the late stage of differentiation.

Subsequently, we identified the expression of genes related to apoptosis. The RT-qPCR results showed that the mRNA expression of the apoptosis-negative regulator *Bcl2* was significantly increased in the miR-130b mimic group compared with the mNC group (*p* < 0.01). While the Western blot results showed that miR-130b mimic treatment tended to decrease the protein expression of BCL2 compared with the mNC group (*p* = 0.064), that of CASP3, an essential executor in apoptosis, was significantly increased (*p* < 0.01) (Figure 7E,F). These results suggest that miR-130b overexpression could induce apoptosis in rat IMAs by inhibiting the post-transcriptional translation of *Bcl2* and further promoting the expression of *Casp3*.

To investigate the mechanism underlying miR-130b inhibiting lipid deposition in rat IMAs, we further detected the expression of key genes related to lipid accumulation in rat IMAs. The results showed that compared with the mNC group, miR-130b mimic treatment significantly increased the mRNA expression of lipoprotein lipase (*Lpl*) (*p* < 0.05) and nuclear receptor subfamily 3 group C member 1 (*Nr3c1*) (*p* < 0.01) and tended to lower mRNA expression of hydroxysteroid 11-beta dehydrogenase 1 (*Hsd11b1*) (*p* = 0.086). Western blot was used to detect the expression of key proteins for lipogenesis, but there was no significant change in PPARG or SREBF1 protein expression between the two groups (Figure 7G,H). These above results indicate that despite promoting apoptosis, increased lipolysis and inhibited adipogenic differentiation were also involved in miR-130b inhibiting lipid deposition in rat IMAs at the late stage of differentiation.

## 3. Discussion

In livestock and poultry, IMF is a crucial component of muscle tissue and a key indicator of meat quality [19]. At the cellular level, IMF content is primarily decided by the number and size of adipocytes, which are determined by proliferation of preadipocytes and the adipogenic differentiation of preadipocytes into adipocytes, characterized by the significant increase in lipid deposition. Therefore, in this study, both the number and size of adipocytes were considered to comprehensively evaluate the effect of miR-130b on lipid deposition in rat IMAs.

MiRNAs are crucial post-transcriptional regulators widely involved in numerous metabolic processes, especially in adipogenic differentiation and lipid deposition regulation [20,21,22]. In addition, a large number of studies have demonstrated the significant role of miR-130b in cancer [23], intervertebral disc degeneration [24], hepatic and biliary stones [25], inflammation [26], and adipogenic differentiation of preadipocytes [27]. Additionally, miR-130b expression in the epididymal WAT of obese mice was significantly higher than in other tissues of the organism [28]. In our previous studies, we demonstrated both in vitro and in vivo that miR-130b inhibited the backfat thickness of pigs by repressing PPARG expression [14,29], and we found that microvesicle-shuttled miR-130b significantly reduced fat deposition in recipient primary cultured porcine adipocytes via PPARG inhibition [13], and this effect was further verified in an HFD-induced obese mice model [15]. These above results suggest that miR-130b can participate in numerous biological processes and could be considered as a negative regulator of subcutaneous fat deposition. However, until now, most studies have focused on the regulatory role of miR-130b in WAT and tumor tissue. Its effect on IMF deposition and the underlying mechanism remain unclear. Therefore, we constructed an in vitro model with rat IMAs to further investigate the effect of miR-130b on adipogenic differentiation and to determine its molecular mechanisms.

It is well known that at the molecular level, adipocyte differentiation is a complex process regulated by several transcription factors, among which *PPARG* and *SREBF1* are the most important ones for regulating numerous genes involved in lipid biosynthesis. Previous studies have shown that PPARG is a molecular switch that activates de novo lipogenesis and lipid accumulation [30]. SREBF1, a central transcription factor regulating lipid metabolism [31], also enhances the transcriptional regulation of *PPARG* by promoting the formation of PPARG ligands. However, in this study, miR-130b overexpression did not change the expression of PPARG and SREBF1, suggesting that miR-130b might inhibit lipid deposition in rat IMAs without directly regulating PPARG/SREBF1 signaling pathways. LPL is a crucial enzyme responsible for the hydrolysis of TG [32]. In this study, miR-130b overexpression significantly increased the mRNA expression of *Lpl*, suggesting that miR-130b might inhibit lipid deposition in rat IMAs by enhancing lipolysis.

Glucocorticoids (GCs) promote the transformation of preadipocytes into mature adipocytes, leading to adipose tissue proliferation [33], and the sensitivity of adipocytes to GCs positively determines their ability to deposit fat [34]. GC action is mainly mediated through NR3C1, and the high expression of NR3C1 often implies a high sensitivity to GCs [35]. A study discovered that the expression level of NR3C1 in porcine IMAs was significantly lower than that in subcutaneous adipocytes, suggesting that the sensitivity of IMA to GCs is much lower than that of subcutaneous adipocytes [36]. HSD11B1 contributed to increasing the concentration of the active form of GCs in tissues, leading to the activation of NR3C1 [37]. In this experiment, *Hsd11b1* mRNA expression tended to decrease after miR-130b overexpression, while *Nr3c1* expression was significantly increased. Since *Nr3c1* is a candidate of miR-130b, this may be due to down-regulation of the post-transcriptional translation of *Nr3c1* by miR-130b overexpression and the subsequent enhancement of the sensitivity of rat IMAs to GCs. However, the exact mechanism still needs further investigation.

IMF content is mainly affected by genetic factors. The increase in the number of adipocytes mainly depends on the proliferation of preadipocytes. Proliferation, differentiation, and apoptosis are the three basic stages of cell development. As an important regulator, miRNA can achieve fine regulation of the cell development process by inhibiting the expression of genes related to proliferation, differentiation, and apoptosis. Numerous previous studies have shown the relationship between miR-130b and apoptosis. In ovarian granulosa cell tumor, miR-130b induced cell proliferation and inhibited apoptosis [38]. In contrast, miR-130b promoted apoptosis in CRC cells, suggesting a potential dual role for miR-130b in apoptosis regulation [18]. However, the role and mechanism of miR-130b in the proliferation and apoptosis of rat IMAs remain to be further studied. Apoptosis can be divided into caspase-dependent and non-caspase-dependent pathways. CASP3 is a key protein in the initiation of caspase-dependent apoptosis. In this study, we showed that miR-130b overexpression significantly promoted the protein expression of CASP3, suggesting that miR-130b reduced lipid deposition by promoting apoptosis in rat IMAs. Furthermore, BCL2, an apoptosis inhibitor, is involved in the regulation of endogenous apoptotic pathways [39]. In this study, we found that although miR-130b overexpression significantly increased the mRNA expression of *Bcl2*, it tended to decrease the protein level of BCL2, indicating that miR-130b promoted apoptosis by inhibiting BCL2, which in turn inhibited intracellular lipid deposition.

A large number of previous studies have shown that miRNA can not only regulate the adipogenic differentiation of preadipocytes but also regulate the proliferation of adipocytes. A previous study showed that miR-10a-5p can inhibit the proliferation and differentiation of chicken myoblasts and promote apoptosis by targeting *Bcl6* [40]. Furthermore, in 3T3-L1 cells, miR-193a-5p overexpression significantly inhibited the proliferation of 3T3-L1 cells by reducing the expression of CDK4 and CCNDB and inhibited the expression of key differentiation markers PPARG, CEBPA, and ACAA2, which finally reduced intracellular lipid deposition [41]. In this study, we showed that miR-130b overexpression significantly inhibited the proliferation of rat IMAs at 24 h and 48 h of differentiation, suggesting that miR-130b can reduce the number of rat IMAs by promoting apoptosis and inhibiting proliferation, thereby reducing lipid deposition in rat IMAs.

The inhibition of cell proliferation is usually caused by the abnormal expression of the cell cycle, so we further explored the expression of cell cycle-related genes. Numerous previous studies have shown that miRNAs can directly or indirectly affect cell cycle progression by regulating key cell cycle gene expressions [42]. Abnormal expression of miRNA can cause cell cycle disorder and lead to the occurrence of a variety of diseases. Moreover, studies have shown that miRNAs are important factors regulating the gene expression of the cell cycle in various cells. For example, miR-322-5p expression was significantly upregulated in damaged liver tissues, while in the mouse hepatoma cell line Hepa1-6, miR-322-5p inhibition significantly inhibited proliferation and apoptosis and blocked the G2/M cell cycle [43]. Previous studies have found that overexpression of miR-130b can promote the proliferation of bovine granulosa cells, cumulus cells [44], and gastric cancer cells [17] and also induce cell cycle arrest in prostate cancer cells [45]. Moreover, during the cell growth index stage of the cervical cancer cell proliferation process, miR-130b can promote cell proliferation, intracellular DNA synthesis, and cell transition from the G1 to S phase. However, in the quiescent phase, miR-130b can inhibit cell proliferation, resulting in increased DNA fragmentation and an increased apoptosis rate [46], suggesting that miR-130b has multiple roles in regulating cell proliferation. In this study, we found that the miR-130b mimic delayed the S+G2/M cell cycle and inhibited cell proliferation of rat IMAs. As far as we know, we demonstrated the first evidence of cell cycle arrest induced by miR-130b overexpression, which might account for its inhibitory effect on lipid accumulation.

In this study, expression of the anti-apoptotic protein BCL2 was significantly reduced in rat IMAs with miR-130b overexpression, while miR-130b inhibition presented the opposite result, suggesting that miR-130b promoted cell apoptosis by inhibiting BCL2 expression. However, through bioinformatics software prediction, we found no targeting relationship between miR-130b and *Bcl2*. Therefore, miR-130b may promote apoptosis by indirectly reducing the expression of BCL2. However, the intermediate regulators in this process need to be further explored.

Much evidence has shown that BCL2 and CASP3 are two key regulatory factors in the regulation of the apoptosis signaling pathway. In this study, we showed that miR-130b triggered apoptosis in the early stage of adipogenic differentiation (24 h and 48 h) of rat IMAs mainly by inhibiting BCL2 expression. While in the late stage of adipogenic differentiation (7 d), the triggering of apoptosis was dependent on the increased expression of CASP3, suggesting that different signaling pathways were involved in miR-130b induced apoptosis of rat IMAs at different stages of adipogenic differentiation. These above results demonstrate that miR-130b plays an important role in regulating the proliferation, apoptosis, and adipogenic differentiation of rat IMAs. However, why miR-130b promotes apoptosis by different signaling pathways in different stages of adipogenic differentiation needs to be further studied.

## 4. Materials and Methods

### 4.1. Animals and Ethics Statements

SPF healthy male Wistar rats (8 weeks of age) used to separate IMAs were provided by Comparative Medical Center of Yangzhou University. Ethical approval was obtained from the research ethics committee of Yangzhou University (approval no. 202303092; date of approval: 11 March 2023). Throughout the course of the research, conscientious efforts were made to mitigate animal distress and minimize the necessity for sacrifices. The protocols for animal experiments were approved by the Jiangsu Administrative Committee for Laboratory Animals, and all procedures performed and parameters applied within the animal facility complied with the Jiangsu Laboratory Animal Welfare and Ethics guidelines (permission number: SYXK-SU-2007-0005).

### 4.2. Cell Isolation, Culture, and Differentiation

Primary IMAs were isolated from the calf skeletal muscle of the rat. Briefly, after removing the fascia and connective tissues, skeletal muscles were cut into pieces by ophthalmic scissors and then centrifuged at 1000 rpm for 5 min. Collagenase type I (Sigma-Aldrich, St. Louis, MO, USA) was used to digest tissue homogenate in a 37 °C water bath for about 2 h, which was then passed through cell strainers (Biologix, Jinan, China) with pore sizes of 600, 400, 200, 100, and 50 mesh. The cell precipitate was gained via centrifugation at 1000 rpm for 10 min. Subsequently, the precipitates were washed with a DMEM medium (Gibco, Grand Island, NY, USA) containing 10% FBS and seeded in T25 cm^2^ bottles. These cells were maintained in the DMEM medium supplemented with 10% FBS and 1% penicillin/streptomycin (Sigma-Aldrich, St Louis, MO, USA) in an incubator at 37 °C and 5% CO_2_. Because the adherent time of muscle satellite cells exceeded 3 h, after differential adherence for 2 h, the supernatant was removed and replaced with a fresh complete medium to obtain relatively pure preadipocytes. Once the preadipocytes grew to 80% confluence, induction solution I containing 45 mL DMEM, 5 mL FBS, 250 μL insulin (1 mg/mL), 50 μL IBMX (0.1 M), 50 μL dexamethasone (2.5 mM), and 5 μL rosiglitazone (2 mM) was replaced to induce differentiation for 3 d. Then, the induced differentiation solution II containing 45 mL DMEM, 5 mL FBS, and 250 μL insulin (1 mg/mL) was replaced for differentiation maintenance and changed every 2 days for a total of 4 days.

### 4.3. Cell Transfection

When the IMAs reached 70–80% confluence on a six-well plate, they were treated with a complete culture medium without antibiotics for 24 h. Either miR-130b mimics (50 nM) or inhibitor of miR-130b (50 nM) was transfected into cells by using Lipofectamine 2000 (Thermo Scientific, North Ryde, NSW, USA) according to the manufacturer’s instructions. Mimic NC and inhibitor NC were transfected as experimental controls. After 5 h of transfection in a 37 °C constant temperature incubator, induced differentiation solutions I and II were replaced to induce differentiation for 24 h, 48 h, and 7 d. Cells were harvested and subjected to subsequent assays. The miR-130b-related sequences used in this study are listed in Table 1.

### 4.4. Oil Red O Staining Extraction Assay and TG Content Determination

The lipid droplets in adipocytes were stained with oil red O. In brief, after the medium was removed, the adipocytes were rinsed twice gently with PBS and subsequently fixed with a 4% paraformaldehyde solution for 40 min. Then, these cells were washed twice carefully with PBS and stained with 2 mL oil red O working solution (3:2 dilution in distilled water and filtered with filter paper) for 30 min. These cells were then rinsed three times with PBS. The stained adipocytes were visualized under a light microscope, and then the stained lipid droplets were dissolved in isopropanol and quantified by measuring the absorbance at 510 nm with a microplate reader (Thermo Scientific). The main component of lipid droplets, TG content, was also calculated using the TG assay kit according to the manufacturer’s instructions (Applygen, Beijing, China). Microscopes were used to obtain images.

### 4.5. RNA Extraction and Quantitative Real-Time PCR

Total RNA was isolated from cells using a TRIzol reagent (Invitrogen Life Technologies, Carlsbad, CA, USA) according to the manufacturer’s protocol. The integrity and concentration of RNA samples were measured using Nanodrop 2000 (Thermo Scientific). Total RNA was stored at −80 °C in a refrigerator. Reverse transcription of mRNA was performed using HiScript III All-in-one RT SuperMix Perfect for qPCR (Vazyme Biotech, Nanjing, China). The quantitative RT-PCR (RT-qPCR) was performed using SYBR Green Master Mix (Vazyme Biotech, Nanjing, China). Relative levels of mRNA expression were determined using the 2^−ΔΔCt^ method. The primer sequences used in this study are listed in Table 2.

### 4.6. Western Blot Assay

After 48 h and 7 d of transfection, the cells were washed twice with PBS and lysed with 1% PMSF (Beyotime, Shanghai, China) and RIPA Lysis Buffer (Beyotime, China). The mixed solution obtained in the previous step was centrifuged at 4 °C for 5 min at 12,000 rpm. After centrifugation, the supernatant was collected, which was a protein solution. The protein concentration was determined by using the BCA protein assay kit (Beyotime, China). The protein samples were subjected to SDS-PAGE and subsequently transferred to PVDF membranes. Antibodies against PPARG (rabbit polyclonal antibody, 1:1000, AP0686, Bioworld, Dublin, OH, USA), SREBF1 (rabbit polyclonal antibody, 1:1000, 14088-1-AP, Proteintech, Rosemont, IL, USA), CASP3 (rabbit polyclonal antibody, 1:1000, 19677-1-AP, Proteintech), BCL2 (rabbit polyclonal antibody, 1:1000, 26593-1-AP, Proteintech), or actin beta (ACTB) (rabbit polyclonal antibody, 1:1000, AP0060, Bioworld) were incubated with the membranes at 4 °C overnight with constant shaking. Then, the membranes were treated with an HRP-conjugated secondary antibody. Proteins were visualized using a chemiluminescent peroxidase substrate (Beyotime, China).

### 4.7. EdU Staining

After 24 h and 48 h of transfection, the proliferation analysis of rat IMAs was conducted by using the Click-iT EdU Cell Proliferation Kit for Imaging (Thermo Scientific). The percentage of EdU-positive cells was examined by fluorescence microscopy (Olympus, Tokyo, Japan). The images were then quantified by Image J software (National Institutes of Health, Bethesda, MD, USA, version 1.53).

### 4.8. Cell Cycle Analysis

The cell cycle distribution was assessed using the Cell Cycle and Apoptosis Analysis Kit (Beyotime, China). Briefly, rat IMAs were cultured in a six-well culture plate. The cells were harvested after 24 h and 48 h of transfection, fixed with 70% ethanol at 4 °C overnight, and resuspended in 400 μL of staining solution and reacted at 37 °C with light avoidance for 30 min before flow cytometer analysis (BD, Franklin Lakes, NJ, USA).

### 4.9. Apoptosis Detection by Annexin V-FITC Method

The apoptosis rate was detected by flow cytometry with the Annexin V-FITC Apoptosis detection kit (Sigma-Aldrich, St Louis, MO, USA). The cells were stained with 5 μL annexin FITC and 5 μL PI for 10 min at room temperature. Cells were then analyzed by fluorescence-activated cell sorting using a flow cytometer (BD, Franklin Lakes, NJ, USA).

### 4.10. Statistical Analysis

All results are presented as mean ± SEM based on at least three separate experiments. SPSS 26.0 statistical software was used to graph and determine significance. Student’s *t*-test (parametric test, unpaired test) was used to determine the significance between two groups. *p* < 0.05 was considered statistically significant.

## 5. Conclusions

In summary, we firstly demonstrated that rno-miR-130 overexpression significantly reduced lipid deposition in rat IMAs at both the early (24 h and 48 h) and late (7 d) stages of differentiation. Furthermore, rno-miR-130b reduced lipid deposition in rat IMAs by inhibiting proliferation and inducing apoptosis at the early stage of differentiation. At the late stage of differentiation, rno-miR-130b inhibited adipogenesis, promoted lipid hydrolysis, and induced apoptosis, thus significantly reducing lipid deposition in rat IMAs. These results provide new insights for increasing the IMF content by inhibiting miR-130b in the livestock and poultry industries.

## Figures and Tables

**Figure 1 ijms-26-01399-f001:**
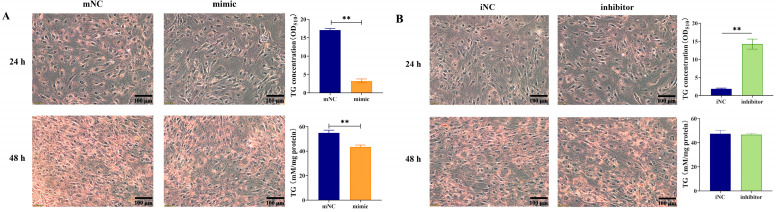
Overexpression of miR-130b significantly inhibited lipid deposition in rat IMAs at early adipogenic differentiation stages (both 24 h and 48 h). (**A**) The effect of miR-130b overexpression on the early adipogenic differentiation of rat IMAs. (**B**) The effect of miR-130b inhibition on early adipogenic differentiation of rat IMAs. Values are presented as the mean ± SEM. ** Represents *p* < 0.01 between mNC and miR-130b mimic groups or between iNC and miR-130b inhibitor groups.

**Figure 2 ijms-26-01399-f002:**
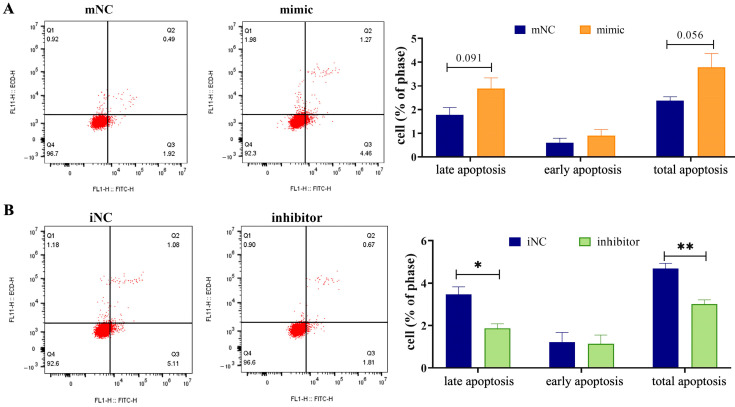
Overexpression of miR-130b significantly promoted the apoptosis of rat IMAs. (**A**) Effect of miR-130b overexpression on apoptosis of rat IMAs after adipogenic differentiation for 48 h. (**B**) Effect of miR-130b inhibition on apoptosis of rat IMAs after adipogenic differentiation for 48 h. Values are presented as the mean ± SEM. * Represents *p* < 0.05, ** Represents *p* < 0.01 between iNC and miR-130b inhibitor groups.

**Figure 3 ijms-26-01399-f003:**
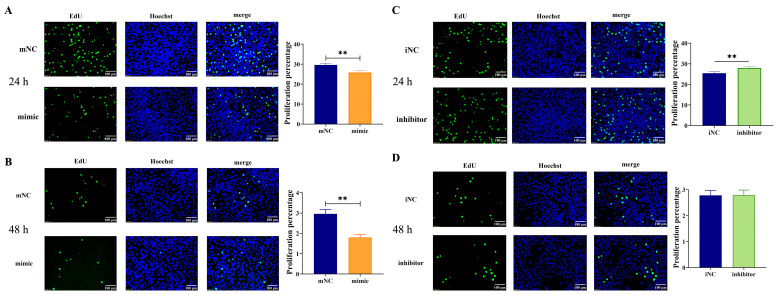
Overexpression of miR-130b significantly inhibited the proliferation of rat IMAs. (**A**) Effect of miR-130b overexpression on proliferation of rat IMAs after adipogenic differentiation for 24 h. (**B**) Effect of miR-130b overexpression on proliferation of rat IMAs after adipogenic differentiation for 48 h. (**C**) Effect of miR-130b inhibition on proliferation of rat IMAs after adipogenic differentiation for 24 h. (**D**) Effect of miR-130b inhibition on proliferation of rat IMAs after adipogenic differentiation for 48 h. Values are presented as the mean ± SEM. ** Represents *p* < 0.01 between mNC and miR-130b mimic groups or between iNC and miR-130b inhibitor groups.

**Figure 4 ijms-26-01399-f004:**
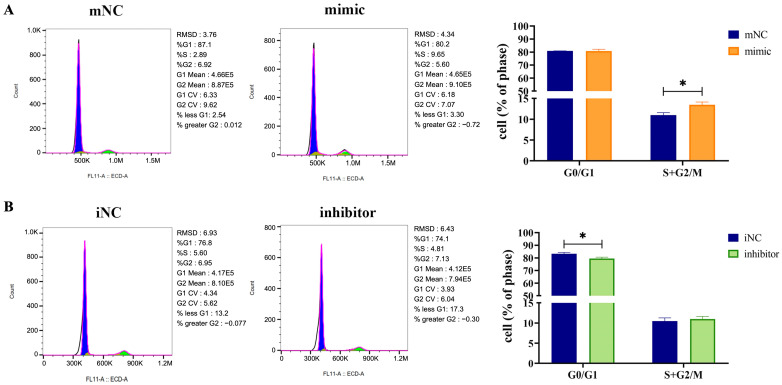
Overexpression of miR-130b arrested the normal cell cycle changes in rat IMAs. (**A**) The effect of miR-130b overexpression on cell cycle change in rat IMAs. (**B**) The effect of miR-130b inhibition on cell cycle change in rat IMAs. Blue represents phase G0/G1 and green represents phase S+G2/M. Values are presented as the mean ± SEM. * Represents *p* < 0.05 between mNC and miR-130b mimic groups or between iNC and miR-130b inhibitor groups.

**Figure 5 ijms-26-01399-f005:**
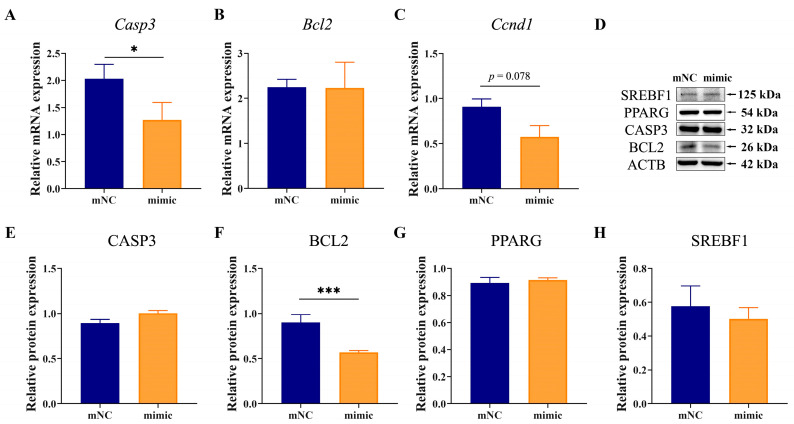
Effects of miR-130b overexpression on expression of factors related to proliferation, adipogenic differentiation, and apoptosis in rat IMAs after adipogenic differentiation for 48 h. (**A**–**C**) *Casp3*, *Bcl2*, and *Ccnd1* mRNA expression. (**D**–**H**) Protein level of CASP3, BCL2, PPARG, and SREBF1. Values are presented as the mean ± SEM. * Represents *p* < 0.05, *** Represents *p* < 0.001 between mNC and miR-130b mimic groups.

**Figure 6 ijms-26-01399-f006:**
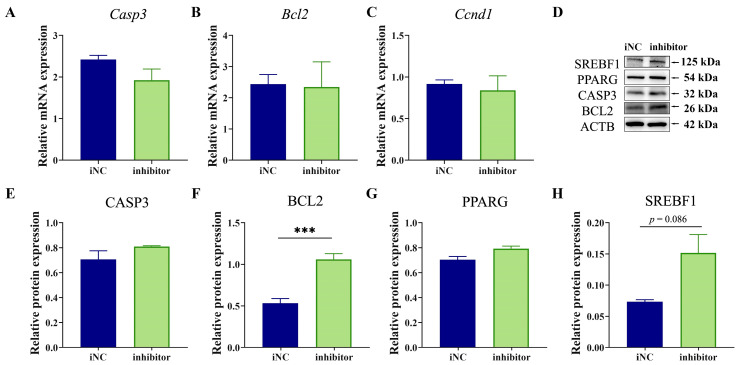
Effects of miR-130b inhibition on expression of factors related to proliferation, adipogenic differentiation, and apoptosis in rat IMAs after adipogenic differentiation for 48 h. (**A**–**C**) *Casp3*, *Bcl2*, and *Ccnd1* mRNA expression. (**D**–**H**) Protein levels of CASP3, BCL2, PPARG, and SREBF1. Values are presented as the mean ± SEM. *** Represents *p* < 0.001 between iNC and miR-130b inhibitor groups.

**Figure 7 ijms-26-01399-f007:**
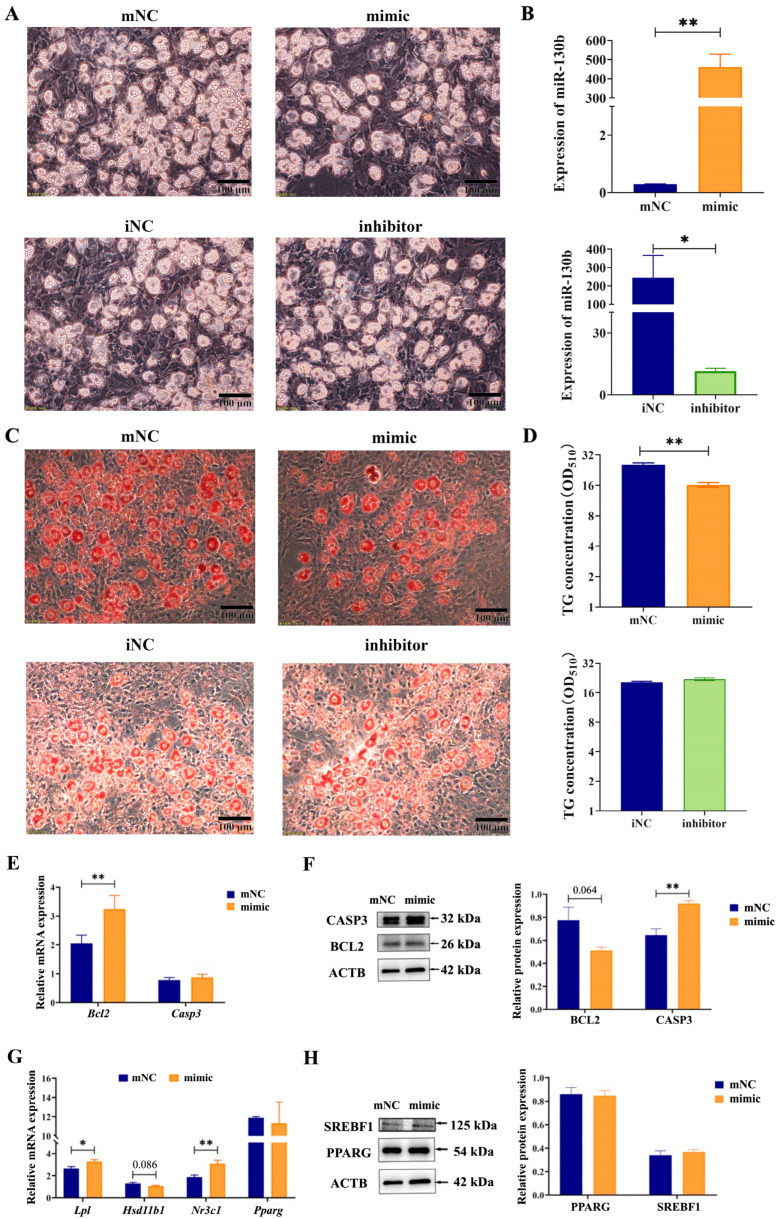
Effect of miR-130b overexpression on adipogenic differentiation of rat IMAs and the underlying mechanism. (**A**) Lipid droplet accumulation in IMAs under the microscope (100 μm). (**B**) Expression of miR-130b in IMAs. (**C**) Oil red O staining of lipid droplets in IMAs (100 μm). (**D**) The quantification of intracellular TG content by measuring the absorbance at 510 nm. (**E**) mRNA expression of key apoptotic genes *Bcl2* and *Casp3*. (**F**) Protein expression of apoptotic key factors BCL2 and CASP3. (**G**) mRNA expression of key adipogenic genes *Lpl*, *Hsd11b1*, *Nr3c1*, and *Pparg*. (**H**) Protein expression of adipogenic key factors PPARG and SREBF1. Values are presented as the mean ± SEM. * Represents *p* < 0.05, ** Represents *p* < 0.01 between mNC and miR-130b mimic groups or between iNC and miR-130b inhibitor groups.

**Table 1 ijms-26-01399-t001:** MiR-130b-related sequences.

**Name**	**Sequence (5′-3′)**
miR-130b mimic	CAGTGCAATGATGAAAGGGCAT
	GCCCTTTCATCATTGCACTGTT
miR-130b mimic negative control	TTCTCCGAACGTGTCACGTTT
	ACGTGACACGTTCGGAGAATT
miR-130b inhibitor	ATGCCCTTTCATCATTGCACTG
miR-130b inhibitor negative control	CAGTACTTTTGTGTAGTACAA
Universal primer	TAGAGTGAGTGTAGCGAGCA
Poly (T) adapter	TAGAGTGAGTGTAGCGAGCACAGAATTAATACGACTCACTATAGGTTTTTTTTTTTTTTTTVN

**Table 2 ijms-26-01399-t002:** Primers used in the present study.

Name	Primer Sequences
*Pparg* (NM_013124.3)	F: TTGATTTCTCCAGCATTTC
	R: TGATCGCACTTTGGTATT
*Lpl* (NM_012598.2)	F: TTGTCCCACTCCGTATCTG R: TATGGTTATCAAGCTCCC
*Nr3c1* (NM_012576.2)	F: AACGTCTGCAACTGGGTC
	R: TGCTTTGGTCTGTGGGATA
*Hsd11b1* (NM_017080.2)	F: GGTGTCTCGCTGCCTTGA
	R: TTCTTCGCACAGAGTGGATA
*Bcl2* (NM_016993.2)	F: CGGGAGAACAGGGTATGA
	R: CTTCATCTCCAGTATCCCAC
*Casp3* (NM_012922.2)	F: CTGGACTGCGGTATTGAG
	R: GGGTGCGCTAGAGTAAGC
*Srebf1* (NM_001276707.1)	F: CACTTACGGTCAGCACTT
	R: CACAACTCACTGGACTTAGA
*Ccnd1* (NM_171992.5)	F: GCGAGGAGCAGAAGTGCGAAGA
	R: GGCGGATAGAGTTGTCAGTGTAGATG
*Gapdh* (NM_017008.4)	F: CCTGGAGAAACCTGCCAAG
	R: CACAGGAGACAACCTGGTCC

## Data Availability

The datasets analyzed during the current study are available from the corresponding author on reasonable request.

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
