# Peer review of "Rno-miR-130b Attenuates Lipid Accumulation Through Promoting Apoptosis and Inhibiting Differentiation in Rat Intramuscular Adipocytes"

_ijms, 2025, doi:10.3390/ijms26041399_

Round 1
Reviewer 1 Report
Comments and Suggestions for Authors
The manuscript entitled “Rno-miR-130b attenuates lipid accumulation through promoting apoptosis and inhibiting differentiation in rat intramuscular adipocytes” explores the impact of Rno-miR-130b on the metabolism of intramuscular adipocytes. While the manuscript presents an interesting and valuable contribution to the field, I have a few suggestions that need to be addressed before publication.
Suggestions for Improvement
1. Nomenclature
All gene and protein symbols should follow the standard nomenclature rules for the respective species. Consistency in nomenclature is crucial to avoid confusion between mRNA and protein expression. For example:
• Humans and non-human primates:
• Full name: peroxisome proliferator-activated receptor γ
• Gene symbol: PPARG (italicized)
• Protein symbol: PPARγ
• Mice and rats:
• Full name: peroxisome proliferator-activated receptor γ
• Gene symbol: Pparg (italicized)
• Protein symbol: PPARγ
Using the same designation for both gene and protein may confuse readers, as they will need to deduce whether the authors refer to mRNA or protein expression. This should be corrected throughout the manuscript.
2. Statistical Analysis
The description of the statistical methods requires clarification. At present, it can only be inferred that the authors used the Student’s t-test for comparisons between two groups and one-way ANOVA for multiple groups. This should be explicitly stated in the Methods section to ensure transparency and reproducibility.
I have no further comments. Congratulations on your interesting research
Reviewer 2 Report
Comments and Suggestions for Authors
In the manuscript ijms-3409776-peer-review-v1, Yu et al. report that Rno-miR-130b attenuates lipid accumulation in rat intramuscular adipocytes (IMAs) by promoting apoptosis and inhibiting differentiation. The authors demonstrate that miR-130b reduces lipid deposition during the early stages of differentiation by suppressing proliferation and inducing apoptosis, and during later stages by inhibiting adipogenic differentiation, promoting lipid hydrolysis, and further enhancing apoptosis. While these findings are potentially interesting, much of the reported data, except for the apoptosis-related results, has already been documented in prior studies. Additionally, the authors have not identified a direct target of miR-130b in the apoptosis pathway, limiting their conclusions to descriptive observations. Given the lack of novel and sufficient data, the claim that "Rno-miR-130b modulates preadipocyte differentiation" does not seem novel. That said, I have no concerns regarding the quality of the presented data.
Round 2
Reviewer 2 Report
Comments and Suggestions for Authors
I agreed with the authors' response.